# Quantitative Evaluation of Groundwater–Surface Water Interactions: Application of Cumulative Exchange Fluxes Method

**Mingqian Li [1,2], Xiujuan Liang [1,2,*], Changlai Xiao [1,2,*] and Yuqing Cao [1,2]**

[1] Key Laboratory of Groundwater Resources and Environment, Ministry of Education, Jilin University, Changchun 130021, China; hydrogeolmq@163.com (M.L.); CAOYQ_hu@126.com (Y.C.)

[2] College of New Energy and Environment, Jilin University, Changchun 130021, China

[*] Correspondence: xjliang@jlu.edu.cn (X.L.); xiaocl@jlu.edu.cn (C.X.); Tel.: +86-1862-691-4139 (X.L.); +86-1350-082-7868 (C.X.)

**Abstract:** Interactions between groundwater and surface water (GW-SW interactions) play a crucial role in the hydrological cycle; thus, the quantification of GW-SW interactions is essential. In this study, a cumulative exchange fluxes method based on mass balance theory is proposed for a stream-aquifer system. This method determines the curve of cumulative fluxes through the water balance term, which can characterize GW-SW interactions, determine the amount of exchange fluxes, and reveal the dynamic process of interactions. This method is used in a reach of the Taizi River Basin, and the GW-SW interactions observed in 2016 are categorized into seven stages and four types (natural controlled, reservoir and irrigation controlled, irrigation controlled, and irrigation hysteresis type). The natural recharge in the study reach is approximately $3.03 \times 10^5$ m³·day⁻¹, and the increase caused by irrigation is $7.8–13.87 \times 10^5$ m³·day⁻¹. After the irrigation stops, the impact can be sustained for 48 d with an increase of $3.03 \times 10^5$ m³·day⁻¹. The most influential factor of the results is the runoff coefficient. The method is applicable to the stream in the plains with upstream and downstream flow monitoring data and can be used to analyze complex GW-SW interactions under the conditions of reservoir storage and agricultural irrigation. The analysis results will provide guidance for the other study of GW-SW interactions in this reach.

**Keywords:** GW-SW interactions; quantification; irrigation; reservoir regulation

## 1. Introduction

When surface water and groundwater act as separate elements in the hydrological cycle, they play significant roles in hydrological and water resource processes. However, with the constant emergence of regional water resource shortages and pollution problems arising from the expansion of human horizons, the study of groundwater–surface water (GW-SW) interactions has gradually gained attention [1,2]. Almost all surface water in nature has a relationship with groundwater, which directly affects the quantity and quality of the water [3–5]. Relevant research of GW-SW interactions plays an important role in the accurate assessment, integrated management, and ecological environmental protection of water resources [2,6,7].

Related studies can be traced back to Boussinesq's discussion of the law of rivers and continuous alluvial aquifers in 1877 [8], which was limited by lack of recognition and did not receive sufficient attention. Since the International Association of Hydrological Sciences (IAHS) and the International Association of Hydrogeologists (IAH) officially labeled GW-SW interactions as an important topic for discussion in 1986 and 1994, the study has become a popular issue in international hydrology and hydrogeology research [9]. Related research mainly includes streams, lakes, coasts, wetlands, springs,

and groundwater systems [1,10–12], among which the stream-aquifer system is the core content of the basin water cycle. Popular research topics include the GW-SW circulation process, water quantity and energy exchange in the hyporheic zone, impact of nature and humanity on the water cycle and water ecology, assimilative capacity of rivers, and instream flow of streams [2,13–15].

The essence of GW-SW interactions is achieved in two steps: the stream infiltrates through sediment into the groundwater (losing stream) and groundwater discharges into the stream (gaining stream). Usually, rivers contain gaining streams in some reaches and losing streams in other reaches. Inside the stream-aquifer systems, GW-SW interactions are dominated by three factors: the characteristics of the aquifer and riverbed, hydraulic gradients, and location and structure of the river, which together determine the amount, direction, and spatial distribution of GW-SW interactions [2]. Outside the system, climatic factors and human activities on water resources indirectly affect the system [3,16]. Therefore, GW-SW interactions involve different dimensions and scales as well as the anisotropic effects of a porous medium, thus exhibiting a certain complexity.

The methods used to estimate the fluxes of GW-SW interactions are as follows. (1) Direct measurements of water flux: Direct measurements across the aquifer-stream-interface are realized by seepage meters; however, the observation of the water flux is at the point scale, making it difficult to obtain data at the surface scale in practical applications. In addition, the measurement may be affected by the resistance of the measurement system itself to flow [17]. Thus, fewer direct measurements are applied at the basin scale. (2) Heat tracer methods: Heat tracer methods extract information using the temperature changes of surface water and groundwater, which are low in cost and suitable for intensive and continuous monitoring. However, this method relies on a significant and stable temperature difference between GW-SW, and daily fluctuations in surface water temperature can also interfere with results [18]. Furthermore, the estimation of flux based on temperature gradients is based on the assumption of vertical flow beneath the surface water [2] and is obtained without the consideration of lateral flow near the riverbank, which directly affects the quantitative results of GW-SW interactions. (3) Methods based on Darcy's law: The groundwater equalization method, hydraulic gradient method, and numerical models are all based on Darcy's law, which depend heavily on the choice of parameters. As the most important parameter, hydraulic conductivity, which is characterized by higher uncertainty, can vary by different orders of magnitude [2,3]. Although the error can be reduced by pumping tests, the anisotropy of the aquifer greatly increases the uncertainty of the method. Numerical simulation is an indispensable tool for water resource management in basins; its accuracy requires the characterization of the formation and hydraulic parameters in a higher degree [19]. (4) Mass balance approaches: Examples are tracer-based methods using isotopic or geochemical tracers of the GW-SW system to reveal the process of GW-SW interactions. These approaches can only yield semi-quantitative results, and they incur high costs, while continuous dynamic monitoring is difficult to achieve. Furthermore, in the area with non-unidirectional interactions between GW-SW, one regional sampling is not sufficient to accurately describe GW-SW interactions [4,20]. The base flow segmentation method can estimate the contribution of groundwater to runoff in precipitation, but this method is based on the assumption that river discharge is directly correlated to groundwater recharge and does not take into account the effects of evaporation or riverbank water storage [2]. In addition, while hydrograph separation works well in mountainous areas, in the plains areas, and especially in areas where water conservancy projects strictly control runoff, the results of segmentation do not represent the true situation. The calculations based on surface water balance are usually based on accrual over months or years; thus, the actual GW-SW interactions cannot be accurately characterized.

No matter which method is used, uncertainty is inherent in the study of GW-SW interactions, which makes it necessary to grasp the basic mode of regional GW-SW interactions in advance when using various methods for research. In practical applications, water balance based on observed streamflow is the most commonly used. However, the water balance usually used on a monthly or annual scale is only applicable to the macro-control of water resources management policy maker and cannot meet the accuracy requirements of responding to GW-SW researches. Calculations on a

daily scale alone cannot accurately reflect the temporal and spatial changes of GW-SW interactions, especially in areas where human activities such as reservoir and irrigation are significant.

Therefore, the objectives of this study were as follows: (1) to propose a cumulative exchange fluxes method based on mass balance to provide an overall framework for corresponding GW-SW researches, (2) to apply this method to a reach of the Taizi River Basin; (3) to verify the accuracy of the method and analyze the errors; (4) to discuss the applicability of the method and its support for other research methods.

## 2. Materials and Methods

### 2.1. Theory of Cumulative Exchange Fluxes Method

Cumulative exchange fluxes is based on the theory of surface water balance, which is the specific performance of the law of mass balance in the hydrological cycle, in which the difference between the input water volume and the output water volume is equal to the volume change of the water body within a certain period. For the main stream, the volume change of the water body can be expressed by the difference between the downstream flow and the upstream flow. The water balance equation is as follows:

$$Q_{gain} - Q_{lose} = \Delta Q = Q_{down} - Q_{up} \tag{1}$$

where $Q_{gain}$ is the sum of the stream recharge, $Q_{lose}$ is the sum of the stream discharge, $\Delta Q$ is the volume change of the stream water body, $Q_{down}$ is the flow at the downstream, and $Q_{up}$ is the flow at the upstream (units of the equation and the following equations are all m³).

Depending on the study area, the recharge and discharge items in the equation are different. The equilibrium exists in most rivers as:

$$Q_t + Q_p + Q_r - Q_e - Q_d + Q_c \pm Q_o = Q_{down} - Q_{up} \tag{2}$$

where $Q_t$ is the flow of tributaries, $Q_p$ is the recharge of precipitation to the stream surface, $Q_r$ is the runoff volume produced by precipitation, $Q_e$ is the evaporation volume of the stream, $Q_d$ is the diversion volume from the river to the outside, $Q_c$ is the volume of the water discharged from the groundwater to the stream, and $Q_o$ represents other water volume changes in the stream.

The usually used monthly data sets of water balance components are difficult to reflect the GW-SW interactions in detail. Therefore, each item in Equation (2) needs daily data sets to support it. Table 1 shows the main fluxes and the methods by which they were quantified. Mean daily flow of upstream and downstream is indispensable. Although the runoff coefficient can be used to roughly estimate the flow of ungauged tributary, the error cannot be ignored. Therefore, when selecting a study reach, it is necessary to avoid rivers with a large number of ungauged tributaries. Due to the unstable flow changes during the flood period, with overbank flooding losses and flood return flows, the error of the calculation increases significantly. Therefore, the period we aim to study should try to avoid the flood period.

**Table 1.** Methods for quantifying water fluxes in the reach water balances.

| Flux and Other Data | Method of Quantification |
|---|---|
| Gauged $Q_{down}$, $Q_{up}$ and $Q_t$ | Daily gauging station data |
| Ungauged, $Q_c$ | Estimated based on runoff coefficients |
| $Q_d$ (for example, canal diversions) | Daily operational gauge measurements |
| $Q_o$ Change in weir storage | Daily operational weir volume measurements |
| River pumping | Daily operational estimates |
| Area of river surface | Simply mean river width multiplied by length or Landsat-based image interpretation |
| Precipitation | Daily mean precipitation from nearest climate station |
| Evaporation | Daily mean evaporation from nearest climate station |

| Runoff coefficient | Estimated from regional hydrological studies that have passed acceptance |
|---|---|

The volume of exchange fluxes $Q_c$ in a certain reach can be expressed as:

$$Q_c = Q_{down} - Q_{up} - (Q_t + Q_p + Q_r - Q_e - Q_d \pm Q_o) \tag{3}$$

For a certain reach, if $Q_c$ is greater than zero in a reach within a certain period, it means that the reach is recharged by the groundwater (gaining stream). Similarly, if $Q_c$ is less than zero within a certain period, it indicates that the reach discharges into the groundwater (losing stream). In the case that the fluxes data referred to in Equation (2) has a daily average value, the volume of the cumulative exchange fluxes for any n days ($Q_{cumulative}^n$) can be obtained as:

$$Q_{cumulative}^n = \sum_{i=1}^n Q_c^i = \sum_{i=1}^n \left[ Q_{down}^i - Q_{up}^i - \left( Q_t^i + Q_p^i + Q_r^i - Q_e^i - Q_d^i \pm Q_o^i \right) \right] \tag{4}$$

where $Q_{item}^i$ is the volume of water of each item on the ith day, and $Q_{cumulative}^n$ is the volume of the cumulative exchange fluxes during the 1st to nth days.

Therefore, the relationship of GW-SW interactions can be judged based on the trend of $Q_{cumulative}^n$. When $Q_{cumulative}^n$ is in a continuously increasing trend for a certain period (Figure 1a,b), it indicates that $Q_c^i$ continues to be greater than zero, that is, the groundwater continues to recharge into the stream (gaining stream). Conversely, when $Q_{cumulative}^n$ is in a downward trend for a long period (Figure 1d,e), this indicates that $Q_c^i$ continues to be less than zero, that is, the stream continues to infiltrate into the groundwater (losing stream).

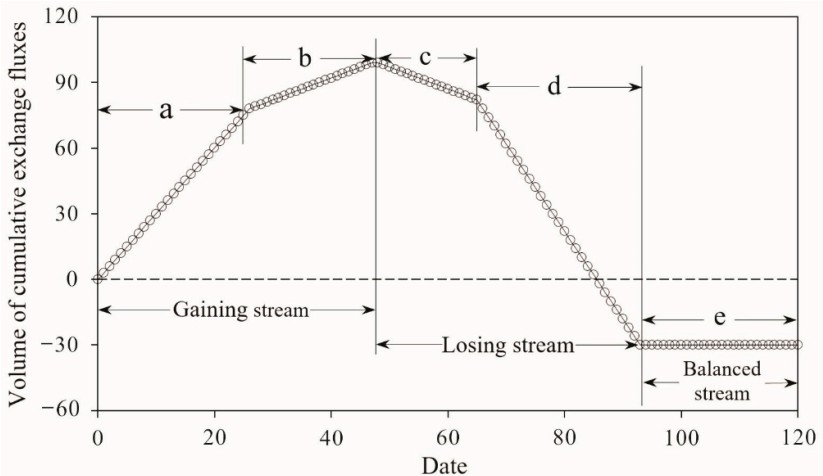

**Figure 1.** Schematic diagram of the results of cumulative exchange fluxes.

In addition, if $Q_{cumulative}^n$ is rapidly increasing (or decreasing) (Figure 1a,d), it indicates that the groundwater is discharged into the stream at a higher rate or that the stream leaks into the groundwater at a higher rate. When the absolute value of the slope of the cumulative curve becomes smaller (Figure 1b,c), the exchange rate decreases accordingly. When the slope is close to zero, it indicates that the exchange between the stream and the groundwater are in equilibrium (balanced stream) (Figure 1e). Therefore, the cumulative exchange fluxes method can be used to judge the dynamic changes and exchange rate of GW-SW interactions and quantify the exchange fluxes over a certain period.

## 2.2. Study Area

The study area belongs to the Taizi River Basin (TRB) in eastern Liaoning Province (Figure 2a), with an area of 13,883 km². The Taizi River originates from the Changbai mountains, and the length of main stream is 413 km, with an average runoff of 37.3 × 10⁸ m³/a. With a warm temperate humid monsoon climate, the precipitation is concentrated in June–August, and the average precipitation is 627.68 mm/a, while the evaporation is 645.08 mm/a. In 1973, the Shenwo reservoir was built in the middle reaches of the Taizi River, located 40 km east of Liaoyang City (Figure 2b), which is a hydro-junction controlled mainly for flood control, irrigation, and urban water supply. The area controlled by the reservoir accounts for 44.5% of the total drainage area, in which the interception and joint scheduling of the reservoir significantly reduces the downstream flow [21].

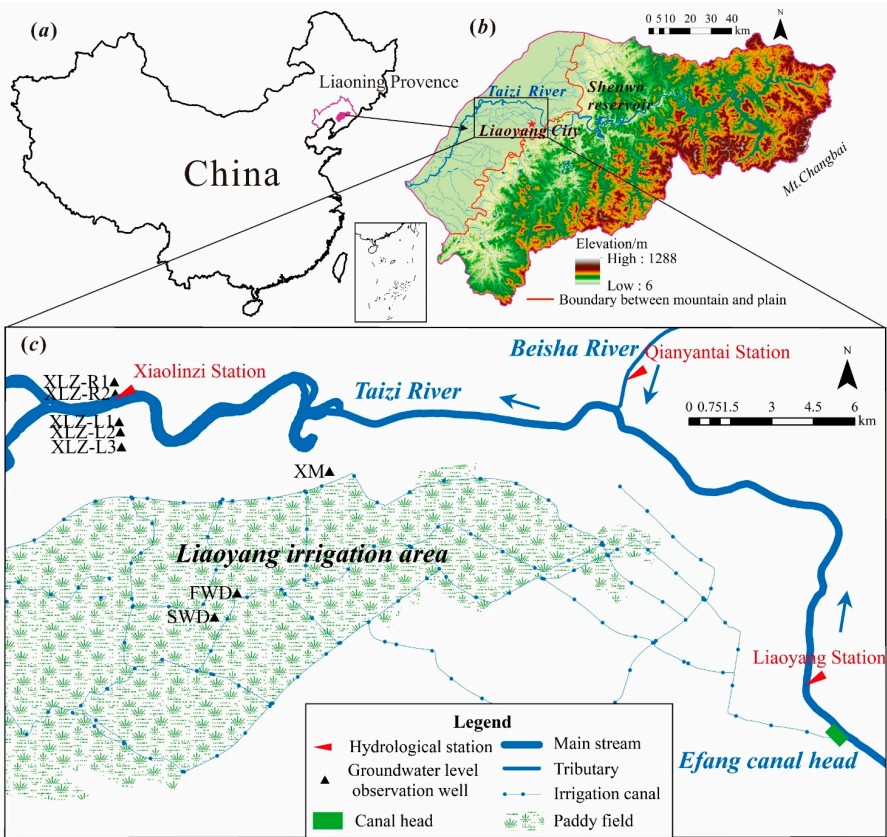

**Figure 2.** Location of the study area (**a**) TRB in China; (**b**) topographic map of the TRB; (**c**) location of the study reach.

The study reach is located in the piedmont alluvial plain of the Taizi River, downstream of the Shenwo reservoir between Liaoyang station, with catchment area of 8082 km², and Xiaolinzi station, with catchment area of 10,254 km², and the length of the reach is 37,727 m (Figure 2c). There is a tributary called the Beisha River, which is monitored by the Qianyantai station with a catchment area of 1112 km². There is no water intake project in this reach. South of the reach is the Liaoyang Irrigation District with an area of 67 km², and the irrigation water comes mainly from the Taizi River through the Efang canal head, which is located in eastern Liaoyang (Figure 2c). Meanwhile, the north of the reach is mainly rainfed cropland. The alluvial plain is flat, and the groundwater occurs in sand and gravel, with a thickness of 40–80 m and a hydraulic conductivity of 70–400 m/day. The aquifer gradually thickens from east to west, with the surface covered by yellow-brown loam of 2–12 m.

### 2.3. Data Collection

The streamflow data sets in 2016 are provided by the Liaoning Provincial Department of Water Resources. The daily average flow data of Liaoyang station, Xiaolinzi station, and Qianyantai station were measured by the vessel-mounted Acoustic Doppler Current Profiler (vessel-mounted ADCP) (HNVF-1000F). The ADCP uses the Doppler-shift phenomenon generated by sound waves, and the velocity is measured by detecting the Doppler shift generated by the sound wave emitted from the moving source at a fixed point [22].

The daily precipitation data were measured by siphonic rain gauge (SL-1) and observed by humans. The cumulative precipitation of two periods, from 20:00 to 8:00 and from 8:00 to 20:00 were measured every day. When the cumulative precipitation was 0.05 mm or greater, the precipitation was measured. The daily evaporation data was measured by a $\Phi$20 cm evaporator (HY.AM3) and observed by humans. The remaining evaporation was observed at 20:00 every day; then, the evaporator was emptied, and 20–30 mm clear water was injected and recorded as the original column of the next day.

The daily average stream level of the Liaoyang and Xiaolinzi stations was observed by humans. The groundwater levels at points XM, FWD, and SWD are represented by the five-day average. The water level at XLZ was taken daily in the rainy season (June–September), and a five-day average was obtained for the rest of the time.

## 3. Results and Discussion

### 3.1. Calculation of the Exchange Fluxes in 2016

According to the method described in Section 2.1., combined with the actual situation of the study reach (recharge of the stream is composed of upstream flow, tributary flow, and the direct and runoff recharges of precipitation, and the discharge is composed of evaporation and downstream flow), the equilibrium equation can be obtained as:

$$Q^{366}_{cumulative} = \sum_{i=1}^{366} Q^i_c = \sum_{i=1}^{366} \left[ Q^i_{down} - Q^i_{up} - \left( Q^i_t + Q^i_p + Q^i_r - Q^i_e \right) \right] \tag{5}$$

where $Q^i_{down}$ refers to the mean daily flow of Xiaolinzi station on the ith day. Similarly, $Q^i_{up}$ and $Q^i_t$ refer to the flow in Liaoyang station and Qianyantai station, respectively. All the flow data were measured by vessel-mounted ADCP.

Precipitation recharge ($Q^i_p$, m³) refers to the amount of water directly supplied by the precipitation on the river surface. As only the daily rainfall data of Xiaolinzi station is collected, and with consideration the reach scale of the study area, the precipitation of Xiaolinzi station can be regarded as the mean precipitation in the study reach (*P*, mm). The area of river surface ($A_{river}$, km²) is determined using Arcgis platform by supervised classification based on satellite images of the study reach in June 2016, which is 5.09 km². The calculation of $Q^i_p$ is:

$$Q^i_p = A_{river} \times P \times 1000 \tag{6}$$

Similar to precipitation recharge, river surface evaporation ($Q^i_e$, m³) is also based on the mean daily evaporation of the $\Phi$20 cm evaporator at Xiaolinzi station. For China, the amount of evaporation observed by the *E*601 evaporator is close to that of natural large water bodies [23,24]. Therefore, when calculating $Q^i_e$, the evaporation of the $\Phi$20 cm evaporator ($E_{20}$, mm) needs to be converted into *E*601 type using a conversion coefficient *C*. *C* (Table 2) is selected according to the *Water Resources of Liaoning Province* [25], which is based on the data of simultaneous observation of two evaporators at 27 stations for a total of 480 years in plain area of the Liaoning Province. The equation for river surface evaporation ($Q^i_e$, m³) is:

$$Q^i_e = A_{river} \times E_{20} \times C \times 1000 \tag{7}$$

**Table 2.** The conversion coefficient *C* between *Φ*20 cm and E601 in different months.

| Month | April | May | June | July | August | September | October | November | December–March |
|---|---|---|---|---|---|---|---|---|---|
| Conversion coefficient *C* | 0.56 | 0.56 | 0.57 | 0.59 | 0.60 | 0.63 | 0.58 | 0.57 | 0.5 |

The runoff ($Q_r^i$, m³) refers to the volume of precipitation that appears as runoff within the catchment area ($A_{catchment}$, km²) and is calculated by the average annual runoff coefficient $\alpha$. $A_{catchment}$ refers to the difference between catchment area controlled by Xiaolinzi station and Liaoyang and Qianyantai station, which is 1060 km². According to the *Water Resources of Liaoning* [25], the runoff coefficient of the study reach from 1956 to 2000 was 0.1–0.2, while 0.15 was selected for the calculation. The calculation equation is:

$$Q_r^i = A_{catchment} \times P \times \alpha \times 1000 \tag{8}$$

The upstream and downstream flow, and precipitation of the study reach in 2016 are shown in Figure 3a. In 2016, the downstream flow was greater than the upstream flow (the sum of the flows from the Liaoyang and Qianyantai stations serve as the upstream flow), and occasionally, the opposite situation occurred (marked by the black circle). Therefore, it is preliminarily determined that the GW-SW interactions were mainly gaining stream. In addition, Figure 3a shows the apparent hysteresis of the stream from upstream to downstream.

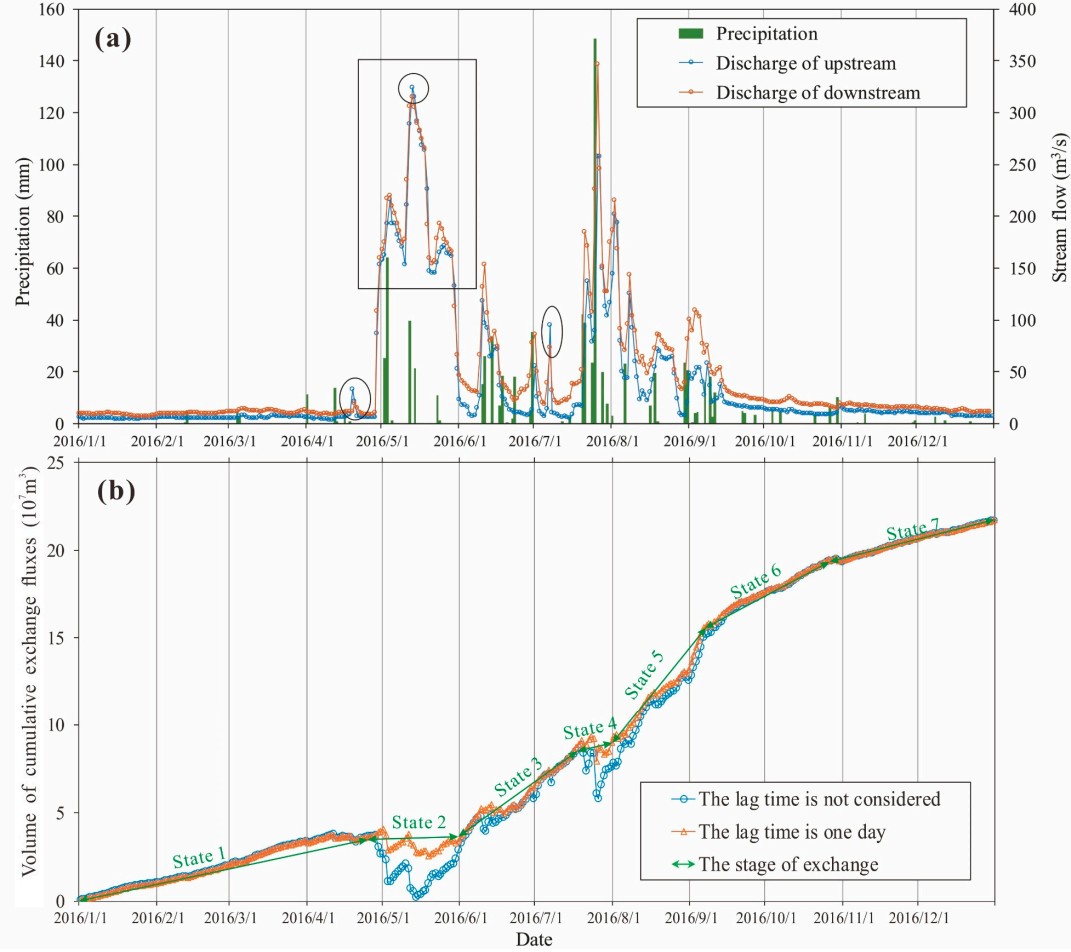

**Figure 3.** (**a**) Precipitation and the flow of upstream and downstream of study reach and (**b**) results of cumulative exchange fluxes.

Hysteresis is a non-linear behavior that is common in natural systems, which is also common in the relation between the streamflow of upstream and downstream. Due to the limitation of the flow velocity and river length, when a certain part of fluxes cannot flow from upstream to downstream within a day, resulting in hysteresis. Therefore, when water balance is calculated on a daily scale, error occurs. In the period of low and stable flow, the error caused by fluxes lag will decrease as the water balance calculation results accumulated and can be negligible. However, when the daily flow becomes large and unstable, the error cannot be ignored. Here we provide a solution. The lag time at the typical flow peak points was counted, as shown in Table 3, with 70% of the flow peak points showing a lag time of one day. Therefore, when calculating the volume of cumulative exchange fluxes, Equation (5) should be corrected to:

$$Q_{cumulative}^{366} = \sum_{i=1}^{366} Q_c^i = \sum_{i=1}^{366} \left[ Q_{down}^i - Q_{up}^{i-1} - \left( Q_t^i + Q_p^i + Q_r^i - Q_e^i \right) \right] \tag{9}$$

**Table 3.** Lag time statistics of typical flow peak in 2016.

| Date in 2016 | 4/19 | 4/30 | 5/13 | 6/10 | 7/1 | 7/22 | 7/26 | 8/2 | 8/8 | 8/19 |
|---|---|---|---|---|---|---|---|---|---|---|
| Peak flow at upstream (m³/s) | 33.52 | 153.57 | 324.3 | 118.91 | 55.4 | 137.8 | 258 | 202.2 | 126.2 | 71.89 |
| Peak flow at downstream (m³/s) | 19.6 | 160 | 315 | 154 | 86 | 185 | 347 | 216 | 144 | 86.5 |
| Lag time (day) | 2 | 1 | 1 | 2 | 1 | 0 | 1 | 1 | 1 | 1 |

The results of cumulative exchange fluxes calculated by Equations (5) and (9) with a lag time of one day and without lag time are shown in Figure 3b. Most of the time, the results are well fitted, and there are deviations only in May and in the late stages of July. As the stream flow is large during this period, the influence of the hysteresis effect increases, and the exchange fluxes shows different results. Taking the stage of the black rectangle (April 27–June 4) in Figure 3 as an example, the flows with a lag time of one day and without lag time are shown in Figure 4a,b, respectively. The comparison results show that a false appearance showing the upstream flow to be greater than the downstream flow will appear without the consideration of the hysteresis effect, which changes the final result. Considering a lag of one day is more representative of the real situation of the flow (Figure 4b), the hysteresis effect can have a false effect on the results of some periods, but the impact on the results for one hydrologic year is minimal. The exchanged fluxes are $21.68 \times 10^7$ m³ and $21.69 \times 10^7$ m³, respectively, without hysteresis consideration and with a lag time of one day. The result with a lag time of one day is considered the final result.

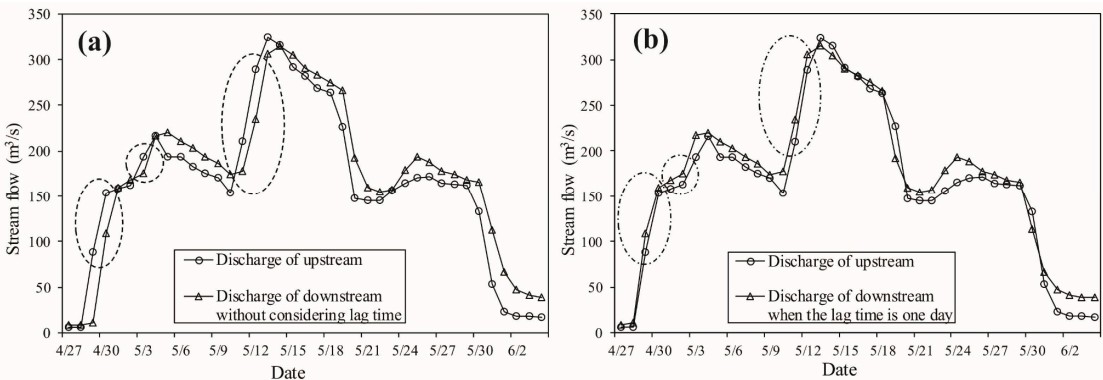

**Figure 4.** The changes in upstream and downstream flow during April 27–June 4. (**a**) Without considering lag time, (**b**) considering the lag time of one day.

### 3.2. Analysis of the Exchange Fluxes in 2016

The amount of cumulative exchange fluxes showed an overall upward trend in 2016, indicating the occurrence of gaining stream. However, the exchange situation is different at different stages,

showing typical stages. The trend of the cumulative curve reveals 7 stages (Figure 3, Table 4), of which the changes in stages 1, 6, and 7 are relatively stable, while stages 2–5 change drastically.

**Table 4.** Results of the volume and rate of exchange fluxes.

| State | Period | Duration Time (day) | Amount of Exchange Fluxes ($10^7$ m³) | GW-SW Interaction | Exchange Rate ($10^5$ m³/day) |
|---|---|---|---|---|---|
| 1 | 1.1–4.27 | 117 | 3.67 | Gaining stream | 3.13 |
| 2 | 4.27–6.2 | 36 | 0.13 | Mainly gaining stream, sometimes losing stream | 0.36 |
| 3 | 6.2–7.19 | 47 | 5.28 | Gaining stream | 11.24 |
| 4 | 7.19–8.5 | 17 | 0.46 | Mainly gaining stream, sometimes losing stream | 2.69 |
| 5 | 8.5–9.13 | 39 | 6.75 | Gaining stream | 17.32 |
| 6 | 9.13–10.31 | 48 | 3.11 | Gaining stream | 6.48 |
| 7 | 10.31–12.31 | 61 | 2.29 | Gaining stream | 3.76 |
| Sum | 1.1–12.31 | 365 | 21.69 | Gaining stream | 5.94 |

Stage 1, from January to April, showed stable gaining stream. During this period, there was less precipitation and no large groundwater exploitation, which caused the groundwater level in irrigation area to decline (Figure 5). The SWD and FWD in the center of the irrigation area began to rise on April 1, contemporaneous with precipitation; thus, precipitation recharge may have been the cause. However, the XM, near the Taizi River, has always been in a downward trend, which may due to the large exchange intensity between the groundwater and the surface water, such that the recharge of precipitation is insufficient to compensate for the discharge into the stream. The upstream and downstream flow is stable, and the volume of cumulative exchange fluxes changes stably (Figure 3b). The exchange rate was $3.13 \times 10^5$ m³/day in stage 1, which can be regarded as the rate of gaining stream under natural conditions. This stage can be considered as natural-controlled type.

River flow increased sharply at stage 2 on April 29 (Figure 3a), without any significant precipitation before, indicating that the steep increase in flow was caused by the draining of the Shenwo Reservoir, which caused the river level to rise rapidly (Figure 5). On the other hand, the Liaoyang Irrigation District began to enter the field soaking period in May, and the Taizi River was drained to irrigate from the Efang canal head. A large amount of stream was recharged into the groundwater, causing the groundwater level to rise rapidly (Figure 5). The river and the groundwater level changed rapidly at this stage, causing fluctuation in the exchange fluxes (Figure 3b) and stream recharge into the groundwater. However, stage 2 still exhibited the gaining stream scenario as a hole, with a very low exchange rate of $0.36 \times 10^5$ m³/day. This stage can be considered as a reservoir- and irrigation-controlled type.

In stage 3, the interstitial release of water from the reservoir, with the decrease in the discharge flow, caused the river level to fluctuate and decrease significantly compared with stage 2. The rice was in the tillering stage, with a decrease in the irrigation intensity (irrigation interval is 8 d with duration of 8 d), causing the groundwater level to fluctuate and slightly increase (Figure 5). Therefore, compared with stage 2, the difference between the groundwater and stream level increased, and the exchange rate rapidly increased to $11.24 \times 10^5$ m³/day. This stage can be considered as irrigation-controlled type.

In stage 4, due to the increase in water discharge from the reservoir, the stream level increased rapidly again, which weakened the recharge intensity gaining stream caused by irrigation, and the exchange rate decreased to $2.69 \times 10^5$ m³/day. This stage is similar to stage 2, which was controlled by reservoir and irrigation together. However, the reservoir discharge duration is not long; thus, this stage lasted for a short time.

Stage 5 is an irrigation-controlled type stage, similar to Stage 3. The groundwater level reached its peak at this stage, and the water-level difference increased (Figure 5), which caused the exchange rate to reach its peak at $17.32 \times 10^5$ m³/day.

In stage 6, the Shenwo reservoir no longer performed large-scale water release, as the irrigation of the paddy fields had ceased. The river and groundwater level were both in a downward trend.

The rate of groundwater recharge into the river water was $6.48 \times 10^5$ m³/day, indicating that although the irrigation was stopped, the influence of previous irrigation still existed, and the groundwater level was still higher than the natural state. Therefore, the exchange rate was still greater than that in the natural state, and this stage can be considered as irrigation-hysteresis type.

The flow in stage 7 gradually recovered to that in stage 1, as the impact of irrigation basically disappeared. The exchange rate was $3.76 \times 10^5$ m³/day, which was close to that in stage 1. Therefore, this state could be considered as natural-controlled type.

In summary, the GW-SW interactions in the study reach were mainly gaining stream, and the recharge rate in the natural state was approximately $3.45 \times 10^5$ m³/day (average of stages 1 and 7). The draining of the reservoir can weaken the recharge capacity of gaining stream and even make the stream discharge into the groundwater. Irrigation enhances the recharge of groundwater into the stream; thus, the increased amount should be attributed to the Taizi River water coming through the Efang channel head. Some of these waters are lost to the atmosphere, some are absorbed by the crops, and the rest are returned to the Taizi River in the form of seepage. The additional exchange fluxes caused by irrigation can reach $7.8–13.87 \times 10^5$ m³/day, which is 2.26–4.02 times that of natural recharge. After the irrigation was stopped, its influence still existed, which slightly enhanced the seepage with a duration of approximately 48 d and an increased recharge rate of approximately $3.03 \times 10^5$ m³/day.

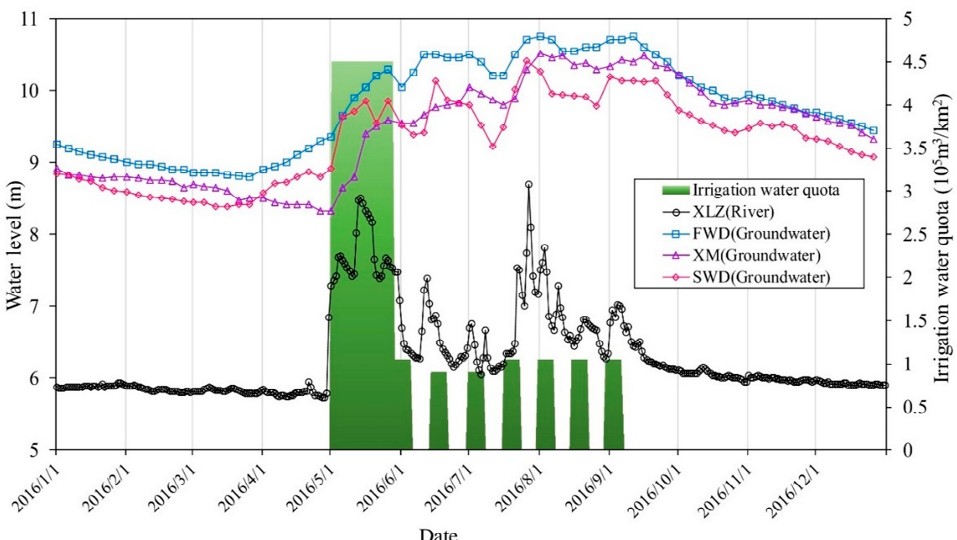

**Figure 5.** Changes in the irrigation water quota and level of the stream and groundwater.

*3.3. Verification of Method's Accuracy*

To verify the correctness of the method, taking the XLZ monitoring section as an example, the changes in the GW-SW level of seven stages are shown in Figure 6. According to the duration of each stage, six levels are uniformly taken to represent the water level change process of the seven stages, simultaneously.

It can be seen from Figure 6 that the changes in the GW-SW level in stages 1 and 7 are slight, reflecting the relatively stable process of gaining stream, which is consistent with the steady increase in cumulative exchange fluxes. The flow in stages 2 and 4 increases instantaneously, and the water level increases greatly, as well (Figure 6), which changes the relationship of the GW-SW interactions from simply gaining stream to alternating gaining and losing stream. In stage 2, the interaction was mainly losing stream, while after the flow was reduced, the interaction was changed back to gaining stream, which is consistent with the results. The water level changes in stages 3, 5, and 6 are relatively small compared to those of stages 2 and 4 (Figure 6). The groundwater level is generally higher than the river level, reflecting the gaining stream, which is consistent with the results in Figure 3.

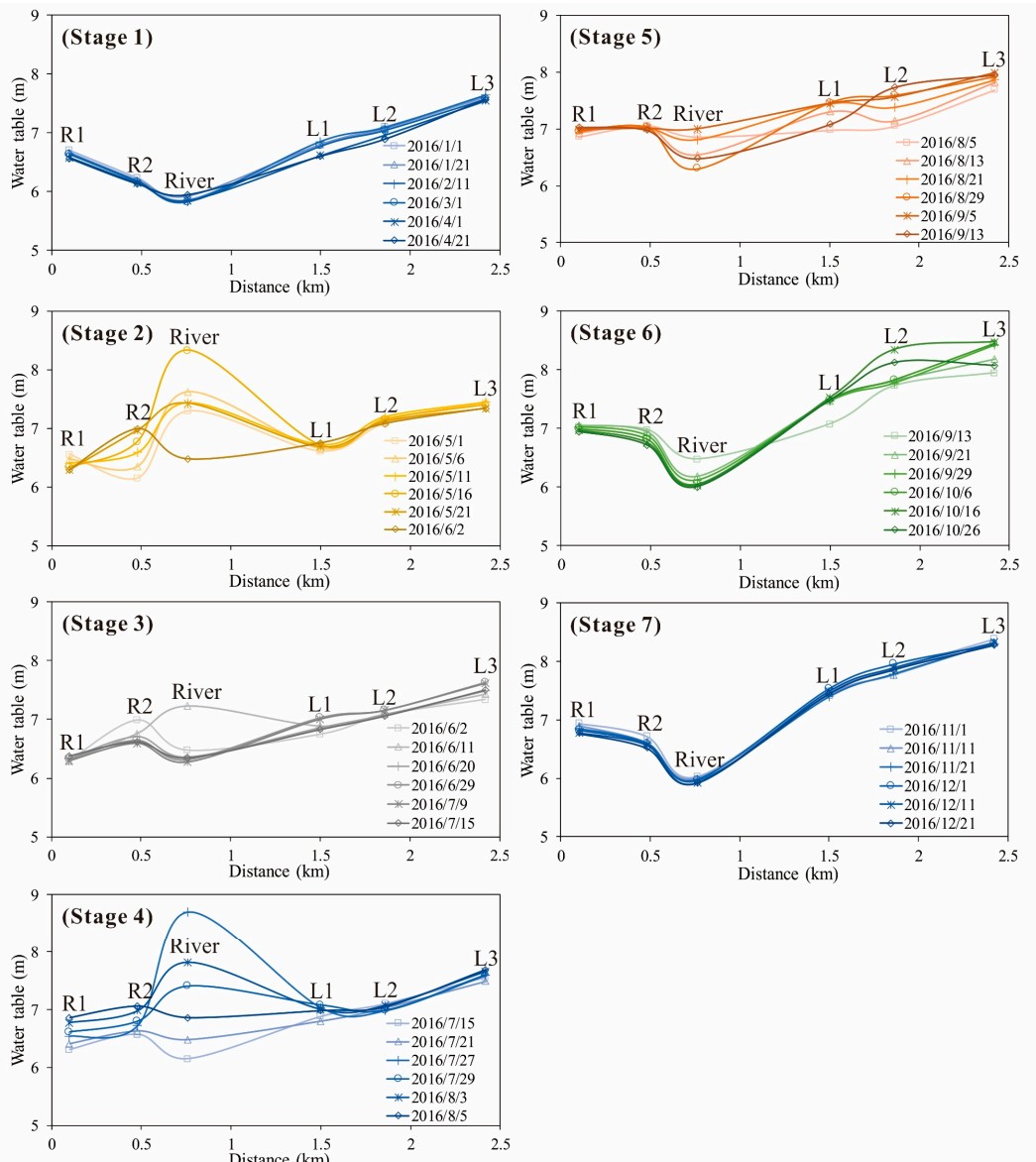

**Figure 6.** Changes in the water level of the XLZ section at different stages (R1 and R2 represent the groundwater monitoring points XLZ-R1 and XLZ-R2 on the right bank, respectively; L1, L2, and L3 represent the XLZ-L1, XLZ-L2, and XLZ-L3 points on the left bank, respectively).

Therefore, the GW-SW interactions reflected by the section is consistent with the result calculated by the cumulative exchange fluxes method. However, in terms of exchange rate, the rate of the section cannot represent the entire reach because of the influence of the topography and the anisotropy of hydrogeological conditions.

### 3.4. Error Analysis of Exchange Fluxes Calculation

According to Equation (9), the factors affecting the result mainly include lag time, measurement error of hydrometeorological data, and runoff coefficient. The effect of lag on the result increases with increasing flow fluctuations, and the lag time is related to the topography of the basin and the length of the study reach, which can be determined according to the typical flow peak point. Hysteresis has little effect on the result of the exchange fluxes in a hydrological year. There exist system errors in the measurements of hydrometeorological data that the error of flow measurement is ±3%, with precipitation of ±0.05 mm and evaporation of ±0.1 mm. As the mass balance is a simple summation,

the error of hydrometeorological data measurement ($\sigma Q_{cumulative}^{366}$) is a linear error propagation and can be obtained as:

$$\begin{cases} \left(\sigma Q_{cumulative}^{366}\right)^2 = \sum_{i=1}^{366}\left[\left(\sigma Q_{down}^{i}\right)^2 + \left(\sigma Q_{up}^{i-1}\right)^2 + \left(\sigma Q_{t}^{i}\right)^2 + \left(k_1 \times \sigma P\right)^2 + \left(k_2 \times \sigma E_{20}\right)^2 + \left(k_3 \times \sigma P\right)^2\right] \\ \sigma Q_{down}^{i} = Q_{down}^{i} \times 3\%, \quad \sigma Q_{up}^{i-1} = Q_{up}^{i-1} \times 3\%, \quad \sigma Q_{t}^{i} = Q_{t}^{i} \times 3\% \\ k_1 = A_{river} \times 1000, \; k_2 = A_{river} \times C \cdot 1000, \; k_3 = A_{catchment} \times \alpha \times 1000 \\ \sigma P = 0.05mm, \sigma E_{20} = 0.1mm \end{cases} \quad (10)$$

The $\sigma Q_{cumulative}^{366}$ is $0.51 \times 10^7$ m³ and the relative errors is ±2.35% compared with the result of 21.69 × 10⁷ m³, which is relatively small.

The runoff coefficient reflects the ratio of precipitation to runoff. This method considers runoff as an equilibrium term, as Equation (9) shows. The average annual runoff coefficient of the study reaching from 1956 to 2000 was between 0.1 and 0.2 [25]. The effect of different values of runoff coefficient on the result is shown in Figure 7. It can be seen that the runoff coefficient mainly changes the amount of exchange fluxes when the precipitation is large. When the precipitation is small, the runoff is primarily not generated, and the impact is basically negligible, such as from January to March and from October to December. In the hydrological year of 2016, when the runoff coefficient was 0.1 and 0.2, the amount of exchange fluxes was 26.31 × 10⁷ m³ and 17.07 × 10⁷ m³, respectively. Compared with 0.15, the relative errors are 21.32% and −21.31%, respectively. It is important to reduce the errors caused by runoff coefficient, especially in non-humid regions. A solution is proposed here. Considering that the runoff coefficient changes greatly during a year, using the monthly-varying runoff coefficients will reduce the error. The variation of the runoff coefficient of the study reach and the corresponding results are shown in Figure 7. It can be seen that the result using monthly-varying runoff coefficients from January to July are basically consistent with result when runoff coefficient is 0.15. The increase in runoff coefficient and precipitation in August made the curve begin to deviate. Compared with 0.15, the relative error is 3.67%. Since the monthly-varying runoff coefficients are closer to the real situation, it is recommended to reduce the error. Although difference exists, the GW-SW interactions reflected by two curves at different stages is consistent. In the absence of monthly-varying runoff coefficient data, the constant runoff coefficient can still be considered.

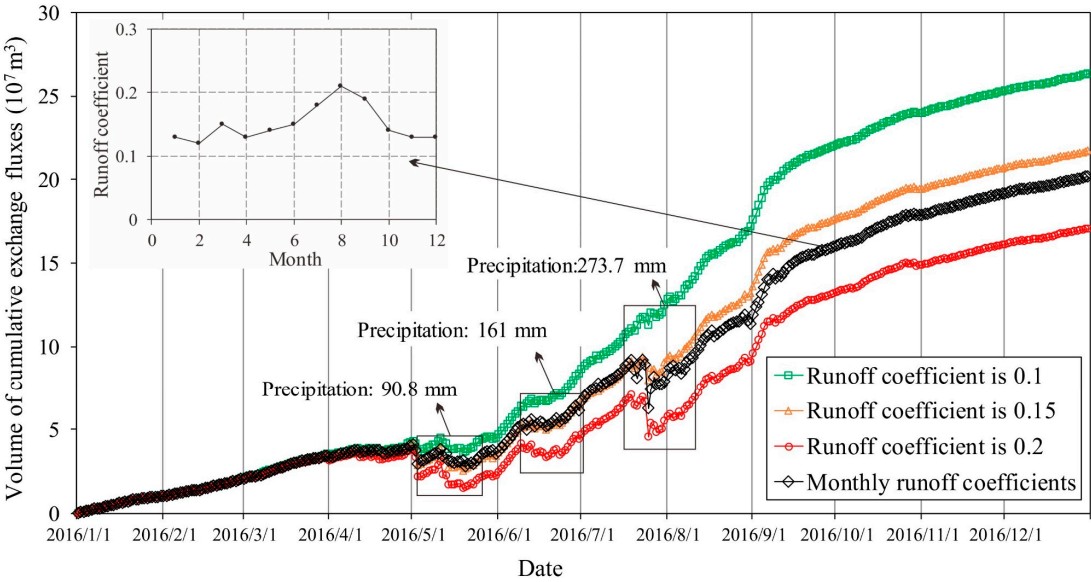

**Figure 7.** Effect of selection of different runoff coefficients on the results of exchange fluxes.

*3.5. The Applicability of Cumulative Exchange Fluxes Method*

This method is based on the principle of surface water balance, so it is applicable to the reach with downstream and upstream flow monitoring data. In addition, the dates of precipitation and evaporation and other equilibrium items that change the stream flow are required. The method is applicable to the study of watershed scale and can also be used to analyze the impact of reservoir regulation and paddy irrigation on GW-SW interactions. For streams with multiple flow monitoring stations, the GW-SW interactions can be studied in a different reach.

As with other methods, this method has certain limitations. First, the study reach cannot be an arbitrary reach; it must be a reach between two flow monitoring points. In addition, the results are mostly affected by the runoff coefficient, which means that for areas with large changes in runoff coefficients, the error increases without sufficient accuracy in the runoff coefficient, e.g., mountains with large changes in topography and areas with large land use changes [26]. However, it is pretty suitable for plains, which have a smaller runoff coefficient. For a reach with a long length, the hysteresis effect of the flow should be considered and can be corrected by the comparison of the typical flow peak points. The calculation results represent the GW-SW interactions of the entire river section, but it is not possible to accurately determine the exchange fluxes of any particular location in the stream, which is also a disadvantage of other methods.

*3.6. Support for other Research Methods*

The cumulative exchange fluxes method can preliminarily determine the relationship and dynamic changes of GW-SW interactions in a reach during a hydrological year, which can provide useful information for research into GW-SW interactions in this reach. For example, when using isotope or hydrochemical tracers, it should be noted that the groundwater source is not the same in different states. When collecting water samples, attention should be paid to separate collection of groundwater, precipitation, irrigation water, and river water. In addition, the exchange rate at different stages is not uniform; thus, water samples collection must be carried out at different states to determine the actual source of exchange fluxes at different stages. Furthermore, the collection can be carried out according to the exchange stages that are presented in Table 4. For the relatively stable stages 1, 6, and 7, the sampling times can be appropriately reduced. For stages 2 and 4 where the GW-SW interactions is more complicated, the number of sampling times can be increased to achieve the maximum benefit of research. In the application of heat tracing, the effect of irrigation on groundwater temperature should be considered. At the same time, the number of measurements should be increased for the complex interactions in stages 2 and 4. This can improve the accuracy of hydrochemical, isotope, and heat tracing methods, and allow the results to more closely approximate real GW-SW interactions. For the GW-SW coupling model at the basin scale, if the study area is divided into a hydrogeological unit, the result of the exchange rate at the relatively stable stage can be converted into an exchange rate of unit length, which can be used to estimate the equivalent hydraulic conductivity of the hydrogeological unit, under the premise of knowing the hydraulic gradient.

## 4. Conclusions

(1) In this study, a cumulative exchange fluxes method based on surface water balance to study GW-SW interactions is proposed. The dynamic change processes of GW-SW interactions can be qualitatively and quantitatively judged through a curve of cumulative exchange fluxes by this method.

(2) The cumulative exchange fluxes method was used to study a reach of the Taizi River in the plains region, and the GW-SW interactions in 2016 can be divided into seven stages and four types (natural controlled, reservoir and irrigation controlled, irrigation controlled, and irrigation hysteresis). The gaining stream rate in the natural state is approximately $3.45 \times 10^5$ m³/day, and the increase in the irrigation due to recharge is $7.8$–$13.87 \times 10^5$ m³/day. After irrigation, the impact can last for 48 d with an increase of $3.03 \times 10^5$ m³/day.

(3) The water-level change of the typical section further confirms the accuracy of the method for determining the GW-SW interactions. The analysis of error shows that the influence of the

evaporation conversion coefficient and flow measurement system is small, the impact of the lag time is small after correction, and the main error comes from the selection of the runoff coefficient.

(4) This method is applicable to reaches in plains with flow monitoring data. When applied areas with large changes in topography or large land use changes, relatively large errors may occur due to the selection of runoff coefficient. The hysteresis of streamflow cannot be ignored when the reach is long. Two solutions have been suggested to deal with selection of runoff coefficient and hysteresis. The basic mode and changes of regional GW-SW interactions can be obtained using this method, but it is not possible to accurately determine the exchange fluxes of any particular location in the stream. Although some limitations exist, this method is effective in analyzing the impact of reservoir regulation and irrigation on GW-SW interactions, and the analysis results have significant implications for studies on other GW-SW interactions carried out in this reach.

**Author Contributions:** M.L., and Y.C. processed the data and analyzed the results; M.L. wrote the manuscript; Y.C., X.L., and C.X. reviewed the manuscript and made helpful suggestions; M.L. revised the manuscript. All authors have read and agreed to the published version of the manuscript.

**Funding:** The research was funded by Natural Science Foundation of China (No. 41572216), the China Geological Survey Shenyang Geological Survey Center "Hydrogeological investigation in the Songnen Plain" project ([2019]DD20190340-W09), the Provincial School Co-construction Project Special -Leading Technology Guide (SXGJQY2017-6), and the Jilin Province Natural Science Foundation (20140101164JC). This work is also partially funded by the Engineering Research Center of Geothermal Resources Development Technology and Equipment, Ministry of Education, Jilin University, China.

**Conflicts of Interest:** The authors declare no conflict of interest.

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
