# Peer review of "Quantitative Evaluation of Groundwater–Surface Water Interactions: Application of Cumulative Exchange Fluxes Method"

_water, doi:10.3390/w12010259_

Round 1

Reviewer 1 Report

The manuscript “Quantitative evaluation of groundwater-surface water interactions, application of cumulative converted flow method” by Li et al. presents an approach to evaluate quantity of groundwater-surface water interaction in a basin scale water system in China. The papers relatively well reviews various pervious methodologies for groundwater-surface water interactions and provides detailed step-wise evaluation of changes in water balance in the river system and thus a useful method that can be used for the water management of the basin. Therefore, it is recommended to be published in Water with a few minor revisions suggested as below.

General comments:

It is not sure if “converted flow” and “conversion” are commonly accepted terminology for the research community for groundwater-surface water interaction. The authors may considered to use “water flux” or other wordings.

This method used in this study is based on water balance between various water volumes within the watershed; thus, the accuracy of each water component, e.g., runoff volume, evaporation volume, etc, is very important The authors should provide more detailed description how each of these components were determined.

Specific comments:

Line 22: Please be cautious about the superscript for the numbers

Line 63: “2) Heat tracer methods” should be bolded.

Line 77: “4) Mass balance approaches” should be bolded.

Lines 91-96: This paragraph should be explained more rigorously: what are advantages and disadvantages of the cumulative converted flow method; information gap from the previous studies; and the specific purpose of this study.

Line 148: Superscript for “8082 km2”.

Line 201: Please provide more information about “hysteresis”. What is this within the context of groundwater-surface water interaction?

Figure 6: Change the axes labeling (e.g., Distance/km à Distance (km)).

Figure 7: Legend order for runoff coefficients of 0.15 and 0.1 should be changed each other.

References: References should be numbered according to the appearance in the main text.

Author Response

Response to Reviewer 1 Comments

 Dear reviewer:

Thank you very much for your comments about our manuscript entitled “Quantitative evaluation of groundwater-surface water interactions, application of cumulative exchange fluxes method".

We have checked the manuscript and revised it according to your comments. We submit here the revised manuscript as well as a list of changes (the replies are highlighted in blue).

General comments 1:

It is not sure if “converted flow” and “conversion” are commonly accepted terminology for the research community for groundwater-surface water interaction. The authors may considered to use “water flux” or other wordings.

Response 1:

We carefully reviewed the literatures on GW-SW interactions again and found that the accepted terminology of “converted flow” and “conversion” should be “exchange fluxes” and “exchange”. We double-checked the relevant words in the manuscript and made a comprehensive change including the text, figures, title and tables.

Where necessary, we still use the word “convert”, but it means something else. For example, “the evaporation of the Φ20cm evaporator needs to be converted into E601 type using a conversion coefficient C”.

General comments 2:

This method used in this study is based on water balance between various water volumes within the watershed; thus, the accuracy of each water component, e.g., runoff volume, evaporation volume, etc., is very important. The authors should provide more detailed description how each of these components were determined.

Response 2:

We have added the extra detailed description of each water component used in water balance, which can be seen in “2.1 Theory of cumulative exchange fluxes method” and “3.1 Calculation of the exchange fluxes in 2016”. In 2.1, we explained how to obtain the fluxes data sets used in water balance and requirements of these data sets in a new paragraph and a new table (Table 1). In 3.1, individual water balance components we used in this case have been explained in more detail, including the sources and processing of all data sets used.

Specific comments 1:

Line 22: Please be cautious about the superscript for the numbers; Line 63: “2) Heat tracer methods” should be bolded; Line 77: “4) Mass balance approaches” should be bolded; Line 148: Superscript for “8082 km2”; Figure 6: Change the axes labelling (e.g., Distance/km à Distance (km)); Figure 7: Legend order for runoff coefficients of 0.15 and 0.1 should be changed each other.

Response 1:

 We really appreciate for the subtle errors the reviewer found! We have carefully revised all the mentioned errors including the subscript, axes labelling, and bold, legend. All changes have been marked in the revised manuscript.

Specific comments 2:

Lines 91-96: This paragraph should be explained more rigorously: what are advantages and disadvantages of the cumulative converted flow method; information gap from the previous studies; and the specific purpose of this study.

 Response 2:

There do were some problems with the original writing of lines 91-96. So, we reorganized the structure of this section and rewritten it. Firstly, the information gap from the previous studies and the significance of the cumulative exchange fluxes method are summarized. Then the specific purposes of this study are briefly descripted. As for advantages and disadvantages of this method, we decided to focus on it in “3.5 The applicability of cumulative exchange fluxes method. And the advantages and disadvantages of this method are also summarized in conclusion (4) which can be found at the end of the manuscript.

Specific comments 3:

Please provide more information about “hysteresis”. What is this within the context of groundwater-surface water interaction?

Response 3:

 We added an explanation of the “hysteresis” in “3.1 Calculation of the exchange fluxes in 2016”. The cause of the “hysteresis” is explained, and the corresponding effects on water balance calculation results under different hydrological regimes on a daily scale are pointed out.

Specific comments 4:

References: References should be numbered according to the appearance in the main text.

Response 4:

 We read the instruction for authors carefully again, checked and modified the format of the reference. It should now meet the requirements of the Water.

Once again, thank you very much for your constructive comments and suggestions which would help us to improve the quality of the paper!

 If you have any question about this manuscript, please don’t hesitate to let us know. Hope these will make it more acceptable for publication. By the way, Merry Christmas!

Sincerely yours,

Dr. Mingqian Li

Corresponding author: Xiujuan Liang

E-mail address: [email protected]

Reviewer 2 Report

As the most influential factor on the results is runoff coefficient (infiltration coefficient) more detailed explanations about it should be done in the paper. It strongly changes during year and this should be taken into account.  Analyses of variation of monthly infiltration coefficients in the study area will be very useful.

The method used in this paper is relatively simple, but problem is that if we wish to have accurate conclusions we need a dense monitoring of  many climatological, hydrological, hydrogeological and other (water use etc.) data. All these factors of the study area, as well as their accuracy should be more detailed explained.

Critical analyses of the method used should be added.

Definitely, the used method is practical and logical but it has a lot of constraints. It should be better explained in the conclusions.

I think that references are not follow the instruction for authors. 

Rows: 63, 77-78 - Heat tracer method, Mass balance method - should be in bold letters.

Author Response

Response to Reviewer 2 Comments

Dear reviewer:

Thank you for your comments on our manuscript entitled “Quantitative evaluation of groundwater-surface water interactions, application of cumulative exchange fluxes method". Your comments are very helpful for revising and improving our paper.

We have studied your comments carefully and made corrections which we hope meet with approval. The main corrections are in the manuscript and the responds to your comments are as follows (the replies are highlighted in blue).

Comments 1:

As the most influential factor on the results is runoff coefficient (infiltration coefficient) more detailed explanations about it should be done in the paper. It strongly changes during year and this should be taken into account.  Analyses of variation of monthly infiltration coefficients in the study area will be very useful.

Response 1:

The variation of monthly runoff coefficients indeed have a significant impact on the results which we didn’t considered before. Therefore, we explained it in more detail in 4.3. And the analyses of monthly-varying runoff coefficients in the study area will minimize the calculation error, leading to a most accurate result. Therefore, we try to calculate the exchange fluxes based on the monthly-varying runoff coefficients. We consulted related research on runoff coefficients in the study area. Fortunately, in Water Resources in Liaoning Provence, there is an analysis of monthly runoff coefficients in the study reach. Using monthly-varying runoff coefficients, new results are calculated, which can be seen in Figure 7, and the corresponding discussion can be seen in 3.4.

Comments 2:

The method used in this paper is relatively simple, but problem is that if we wish to have accurate conclusions we need a dense monitoring of many climatological, hydrological, hydrogeological and other (water use etc.) data. All these factors of the study area, as well as their accuracy should be more detailed explained.

Response 2:

An extra detailed description of each data set used in water balance has been add in “2.1 Theory of cumulative exchange fluxes method”. In 2.1, we explained how to obtain these data sets used in water balance and corresponding requirements in a new paragraph and a new table (Table 1).  As these factor in the study area, in“3.1 Calculation of the exchange fluxes in 2016”, individual water balance components we used have been explained in more detail, including the sources and processing of all data sets used. As for the accuracy of these factors, we decided to focus on it in “3.4 Error analysis of exchange fluxes calculation” where measurement error of all hydrometeorological data and their impact on the results are discussed and calculated.

Comments 3:

Critical analyses of the method used should be added.

Response 3:

In fact, we did not specifically conduct critical analysis separately, but divided the critical analysis into two chapters, 3.4 Error analysis of exchange fluxes calculation and 3.5 The applicability of cumulative exchange fluxes method. In 3.4, we discussed the measurement error of hydrometeorological data in the water balance, and calculated the effect of the measurement error on the results through the linear error propagation formula. In 3.5 we discussed the applicability of this method in different regions, and also pointed out the limitations, advantages and disadvantages (mainly disadvantages) of this method, mainly including: (1) the study reach cannot be an arbitrary reach, it must be a reach between two flow monitoring points; (2) the effects of runoff coefficient and hysteresis cannot be ignored; (3) it is not possible to accurately determine the exchange fluxes of any particular location in the reach.

Comments 4:

Definitely, the used method is practical and logical but it has a lot of constraints. It should be better explained in the conclusions.

Response 4:

We have carefully revised Conclusion (4), which now includes the summarized disadvantages (constraints) and advantages of this method.

Comments 5:

The references are not follow the instruction for authors. 

Response 5:

We read the instruction for authors carefully again, checked and modified the format of all references one by one. References are numbered according to the appearance in the main text now. It should now meet the requirements of the Water.

Comments 6:

Rows: 63, 77-78 - Heat tracer method, Mass balance method - should be in bold letters.

Response 6:

We really appreciate for the subtle format errors the reviewer found and we have revised the mentioned format errors now.

Once again, thank you very much for your constructive comments and suggestions which would help us to improve the quality of the paper!

 If you have any question about this manuscript, please don’t hesitate to let us know. Hope these will make it more acceptable for publication. By the way, Merry Christmas!

Kind regards,

Mingqian Li

 Corresponding author: Xiujuan Liang

E-mail address: [email protected]

Round 2

Reviewer 2 Report

The  corrected version of the paper can be accepted.